# Development and psychometric properties of Iranian midwives job satisfaction instrument (MJSI): A sequential exploratory study

**Ashraf Direkvand-Moghadam[1], Nasrin Rashan[2], Mona Bahmani[3], Safoura Taheri●[1]***

**1** Department of Midwifery, Faculty of Nursing and Midwifery, Ilam University of Medical Sciences, Ilam, Iran, **2** Faculty of Nursing and Midwifery, Ilam University of Medical Sciences, Ilam, Iran, **3** Faculty of Health, Ilam University of Medical Sciences, Ilam, Iran

\* taheri_safura@yahoo.com

## Abstract

### Background

Job satisfaction refers to a person's attitude toward his/her job and its various aspects. Job satisfaction improves the quality of service and employees' physical and mental health. The present study aimed to design a valid and reliable instrument to assess Iranian midwives job satisfaction instrument (MJSI).

### Methods

This is a sequential exploratory study for tool design. This study in two phases; (qualitative and tool's psychometric evaluation) was conducted in Ilam, Iran, 2019 years. In the first phase, a qualitative content analysis was carried out by in-depth and semi-structured individual interviews with 10 experts. Then, the pool of items extracted from the qualitative phase was completed by reviewing the existing texts and tools. The second phase of the study involved reducing the overlapping items and validating the tool. In order to investigate the construct validity, a cross sectional study was conducted with the participation of 121 midwives with census sampling. Data analysis was performed by SPSS-19 software using exploratory factor analysis and reliability tests (Cronbach's alpha).

### Results

In the qualitative phase and after reviewing the existing texts and tools by the research team, a 58-item questionnaire was developed and then entered into the psychometric phase. Then, the tool was finalized with five factors, including: 1) communication features, 2) professional features, 3) responsibility aspects, 4) physical-mental aspects and 5) social aspects, respectively. After the psychometric process, by removing the items in different stages, a specific questionnaire was developed to measure the midwives' job satisfaction with 25 items which explained a total of 49.95% of the total variance. Reliability of the tool was approved by Cronbach's alpha = 0.71 and test-retest with 2-weeks intervals, indicating an appropriate stability for the scale (ICC = 0.898).

**Data Availability Statement:** All relevant data are within the paper and its Supporting information files.

**Funding:** The authors received no specific funding for this work.

**Competing interests:** The authors have declared that no competing interests exist.

## Conclusion

The 25-item self-reporting midwives job satisfaction tool had acceptable validity and reliability. We recommend the use of this tool for evaluating the job satisfaction of midwives, as well as management and research purposes.

## Background

Job satisfaction refers to a person's attitude toward his/her job and its various aspects. It is one of the important issues that service organizations should consider [1]. Certainly, increased job satisfaction improves the quality of service. In addition to improving organizational service, job satisfaction is also linked to employees' physical and mental health and increase to the quality of service [1, 2].

So far, several factors have been identified as influencing factors in job satisfaction [3, 4]. Job satisfaction is more important in the medical professions. Midwives have a special place in the medical profession for several reasons. Firstly, midwives as an important component of health care system, play a key role in delivering services to two vulnerable groups of society such as mothers and children. Secondly, midwifery services are offered on a large scale from prevention in health care to advanced medical services in hospital [5].

Given the importance of mothers and children as the main target groups in midwifery services around the world, it is essential to identify midwives' job satisfaction. Therefore, various studies have been carried out so far and different results have been reported [5, 6]. A high rate of job satisfaction has been reported in Australian midwives. The results of one study reported the 81% job satisfaction among Australian midwives. However, the job satisfaction rate is very low in Iranian midwives (1.6%), [1].

On the other hand, various studies have reported many related factors to job satisfaction. In some studies, welfare and management aspects have been investigated as main factors that affect job satisfaction. In other studies, the job security, income level, and work environment are the most important factors in job satisfaction [7, 8].

The measurement of job satisfaction plays an important role in the promotion of health care systems. Also, variety of factors that affect job satisfaction and their impact level in various countries and cultures highlights the need to design and standardize a tool for measuring satisfaction in different cultures and jobs. In this regard, there has been an efforts to design and standardize job satisfaction assessment tools for Iranian physicians and nurses, but no such tool has been designed for Iranian midwives. Since midwives are one of the most essential components of service delivery to the two vulnerable groups of society (mothers and children), the midwives' job satisfaction is a concern to health service organizations in both developed and developing countries [9]. Considering that, the studies conducted in the field of midwives' job satisfaction have used non-specific tools [10–12]) and also, due to the importance of this issue, we decided to take a qualitative approach and use content analysis to examine the job satisfaction of Iranian midwives and accordingly, develop a questionnaire with psychometric characteristics. It should be noted that the main researchers in this study have experience in instrument design and psychometrics[13, 14]. The present study aimed to design a valid and reliable tool to assess the job satisfaction in Iranian midwives.

## Methods

### Design

A mix method study (sequential exploratory study) was conducted from April to December 2019. This research started with a qualitative study and continued with a quantitative study. This method was introduced by Creswell and Plano Clark as one of the five main types of mix method studies [15]. The sequential exploratory study is divided into two types of theory design and instrument design. The present study is a tool design study [16]. To build a tool, we need to understand the concept of health literacy by using the experience of participants. Having a conceptual framework is the first step in designing a tool. Therefore, the qualitative part of the study was designed and implemented to design the tool, and then its psychometric property was confirmed in a quantitative study.

Both quantitative and qualitative studies were applied studies. To develop a tool, it was necessary to know the concept of job satisfaction through the experiences of study participants, because having a conceptual framework is the first step to building tools. Therefore, the qualitative part of the study was implemented to design a tool, and then in a quantitative study, the psychometric properties such as validity and reliability of the tool were confirmed. the tool's validity was obtained by face validity, content validity and factor analysis methods. The internal consistency and test–retest reliability methods were used to confirm the reliability of the tool.

### Phase 1: Development of job satisfaction tool

In the this phase, a qualitative content analysis was carried out by in-depth and semi-structured individual interviews with 10 experts. The inclusion criteria were; being a faculty member or a midwife with at least five years of experience in health centers, hospitals, and counseling centers who were key informants. The interview was conducted by agreement between the researcher and the participant in any place where the participants could more easily share their experiences. The first question to start interview was: "What comes to mind when you hear the word job satisfaction?. As the study progressed, additional questions or new questions were asked from the interviewees. Exploratory questions such as: "What do you mean?" or "If you can please explain more" or Please explain to me? or Please tell me about your experience were also used in the interviews as needed (The Questions guide in qualitative phase is attached in the S1 File section). At the end of each interview, participants were asked to add if there was anything left to say. The interview was conducted by the researcher. The duration of each interview was between 45 to 110 minutes, depending on the amount of information, participation, and cooperation of the participants.

Qualitative analysis was performed using contract content analysis and MAXQDA-10 software.

After analyzing the findings of qualitative phase and reviewing the existing texts and tools, a pool of item was created. After reviewing the items by the members of research team, the initial format of the midwives' job satisfaction questionnaire with 58 items was formed. The midwives' job satisfaction questionnaire consisting of five sections: 1) communication features, 2) professional features, 3) responsibility aspects, 4) physical -mental aspects and 5) social aspects. Then the research entered the psychometric stage.

### Phase 2: Psychometric evaluation

We used face validity, content validity and factor analysis as the main methods of the tool' validity assessment. Face validity conducted with qualitative and quantitative methods. In the

quantitative face validity, the level of difficulty, relevance and ambiguity of the items were investigated by face to face interview with 10 midwives (faculty member and midwife working in health centers, hospitals, and counseling centers). Then, to determine the quantitative face validity, the impact score method was used. An item with a score equal to or greater than 1.5 was retained and other items were deleted. A 5-point Likert scale was used for determining the importance of items. The classification of very important (5), important (4), relatively important (3), slightly important (2), and unimportant(1) was considered [17]. Therefore, 10 midwives were asked to rate the importance of each item. For content validity, we used both quantitative and qualitative methods, as one of the most important steps of tool's validity. In qualitative content validity, 10 expert members were consulted and asked to carefully study the tool, and their corrective views and suggestions regarding content coverage, grammar, use of appropriate phrases and the appropriate location of the items were considered. For quantitative content validity, we used Content Validity Ratio (CVR) and Content Validity Index (CVI). A 3-point Likert scale was used to determine CVR by an expert panel. The calcification of "not essential, useful but not essential, essential" was used at this stage, so that higher level of content validity showed the agreement between greater number of experts(faculty member and midwife working in health centers, hospitals, and counseling centers) in the experts' panel. In the present study, according to 10 specialists and based on the Lawshe table, if the CVR coefficient was more than 62%, the item was retained.

Then a content validity index (CVI) was computed for the instrument. In this section we asked the expert panel to rate each item in terms of clarity and relevancy in a 4-point ordinal scale (1 = not relevant, 2 = somewhat relevant, 3 = quite relevant, 4 = highly relevant). CVI was calculated for I-CVIs and the scale-level (S-CVI). The I-CVI expressed the proportion of agreement on the relevancy of each item, which was between 0 and 1. If the validity index of each item was higher than 0.79, the validity of that item would be accepted. The exploratory factor analysis was applied to determine the construct validity. So, the questionnaires were completed by 121 midwives with census sampling in a cross-sectional study(A minimal set of data for the study is attached in the S2 File section). Therefore, the inclusion criteria were; all faculty member and all midwife working in health centers, hospitals, and counseling centers that would like to participate in the study, with any level of work experience.

In this study, considering that the number of items in the midwives' job satisfaction questionnaire was 25 items, the sample size for performing exploratory factor analysis was estimated to be 5 samples per item. Kaiser-Meyer-Olkin (KOM) and Cruet-Bartlett's test and principal component analysis, as well as Varimax rotation. The "Eigen Value" and "Scree Plot" were used to determine the number of factors. A minimum of 40% load was required to keep each factor extracted from factor analysis. Eigen Value of more than 1 was considered. We used internal consistency and test–retest reliability methods to confirm the reliability of the questionnaire. In the present study, Cronbach's alpha was performed using SPSS-19 software with a minimum coefficient of 0.7. To determine the stability of the questionnaire, 10 midwives (faculty member and midwife working in health centers, hospitals, and counseling centers). were asked to complete the final questionnaire twice, at an interval of 2 weeks.

## Ethical considerations

This study was carried out with the approval of the Ethical Committee of Ilam University of Medical Sciences (N: 22.52.98.1572). The aim of the study was explained to all participants before their enrollment in the study. All personal data such as names were anonymized. Paper based data (consent) was stored securely in a locked cupboard and electronic data (interview

and transcripts, quantitative data) was stored in secure server with password (The consent form used is attached in the S3 File section).

## Results

In qualitative phase, in total, 10 expert person aged 27–45 years participated in this study. The lowest work experience was five years and the highest work experience was 26 years. The participants' current tasks of the participants were included two supervisors, two staff, two Training, two health service provider, one gynecologist assistant and one clinic staff.

In the present study, we used literature review and interview with the participants in qualitative phase to develop the tool. At this stage, 58 items were obtained, which indicated the midwives' job satisfaction. The participant´s level of attitude was determined using a 5-point Likert scale, ranking from; totally agree = 4, agree = 3, no comment = 2, disagree = 1 and strongly disagree = 0.

In the qualitative and quantitative face validity, 9 items had the item impact of less than 1.5 and were excluded from the questionnaire. In the content validity section, 24 items did not obtain the necessary score based on Lawshe formula, so they were excluded from the questionnaire.

The exploratory factor analysis was used to determine the construct validity. For this purpose, the questionnaires were completed by 121 midwives in a cross-sectional study. Demographic data for the sample are presented in Table 1. The factor analysis was done on 25 items based on principal component analysis. Items' validity was determined by KMO index and

**Table 1. Demographic characteristics and other factors related to job satisfaction in midwives.**

| Variable | Category | N(%) |
|---|---|---|
| Age (years) | Less than 25 | 29(24) |
| | 26–30 | 19(16) |
| | 31–35 | 9(8) |
| | 36–40 | 17(14) |
| | More than 40 | 47(38) |
| | Total | 121(100) |
| Marital status | Single | 35(29) |
| | Married | 85(70.2) |
| | divorced | 1(0.8) |
| | Total | 121(100) |
| Work experience (years) | Less than 5 | 47(38.8) |
| | 6–10 | 9(7.5) |
| | 11–15 | 21(17.3) |
| | More than 16 | 44(36.4) |
| | Total | 121(100) |
| Current task | Supervisor | 3(2.3) |
| | Staff | 10(8.3) |
| | Therapist | 56(46.3) |
| | Training | 8(6.6) |
| | Health service provider | 25(20.7) |
| | Clinic staff | 5(4.1) |
| | Gynecologist assistant | 4(3.3) |
| | Administrative department | 10(8.4) |
| | Total | 121(100) |

**Table 2. Kaiser-Meyer-Olkin Measure and Bartlett's Sphericity test of Sampling Adequacy of the questioner of the job satisfaction among Iranian midwives.**

| KMO and Bartlett's Test | | |
|---|---|---|
| Kaiser-Meyer-Olkin Measure of Sampling Adequacy. | | 0.640 |
| Bartlett's Test of Sphericity | Approx. Chi-Square | 845.696 |
| | Df | 300 |
| | Sig. | .000 |

Bartlett's test, which were carried out on 25 Items. In this study, the value of KMO index was 0.64 (Table 2).

The scree plot and eigenvalue methods were used to determine the number of question-naire's constituents Based on the scree plot, five factors were proposed for extraction in explor-atory factor analysis of Iranian midwives' job satisfaction instrument which are shown in Fig 1. After calculating the correlation matrix between the variables, factor extraction was per-formed by analyzing the main components using orthogonal rotation with eigenvalues >1.

Overall, 9 factors were extracted by factor analysis. However, 49.95% of the total variance was explained by 6 factors, and other factors explained the remaining total variance. Totally, 9.12% of rotation variance was explained by first factor and 8.71% by the second factor (Table 3).

Based on the rotation matrix components, the first factor was composed of 7 items and the second factor was composed of 10 items. The third and fourth factors had 2 and 3 items, respectively. Other factors were composed of 1 item, therefore, this item was transferred to other factors according to the face validity and the opinion of researchers. Eventually, the items were combined into 5 factors, which included communications (7-item, questions 1 to

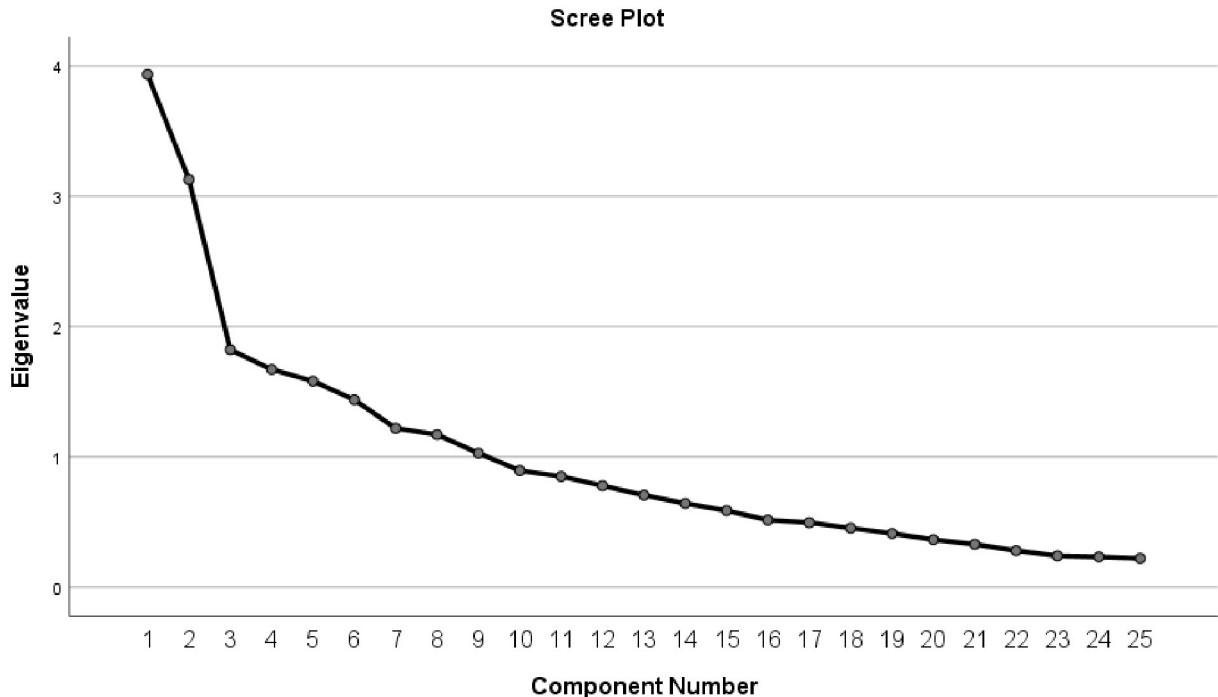

**Fig 1. Scree plot.** Based on the Scree plot, five factors were proposed for extraction in exploratory factor analysis of Iranian midwives' job satisfaction instrument.

**Table 3. The total variance explained by factors of the job satisfaction tool among Iranian midwives.**

| Component | Initial Eigen values | | | Extraction Sums of Squared Loadings | | | Rotation Sums of Squared Loadings | | |
|---|---|---|---|---|---|---|---|---|---|
| | Total | % of Variance | Cumulative % | Total | % of Variance | Cumulative % | Total | % of Variance | Cumulative % |
| 1 | 3.934 | 15.738 | 15.738 | 3.934 | 15.738 | 15.738 | 2.280 | 9.122 | 9.122 |
| 2 | 3.128 | 12.512 | 28.250 | 3.128 | 12.512 | 28.250 | 2.178 | 8.714 | 17.836 |
| 3 | 1.823 | 7.292 | 35.542 | 1.823 | 7.292 | 35.542 | 2.143 | 8.573 | 26.408 |
| 4 | 1.671 | 6.684 | 42.226 | 1.671 | 6.684 | 42.226 | 2.061 | 8.243 | 34.652 |
| 5 | 1.581 | 6.326 | 48.552 | 1.581 | 6.326 | 48.552 | 2.000 | 8.000 | 42.651 |
| 6 | 1.438 | 5.751 | 54.303 | 1.438 | 5.751 | 54.303 | 1.825 | 7.300 | 49.951 |
| 7 | 1.219 | 4.877 | 59.180 | 1.219 | 4.877 | 59.180 | 1.677 | 6.708 | 56.659 |
| 8 | 1.171 | 4.683 | 63.863 | 1.171 | 4.683 | 63.863 | 1.525 | 6.101 | 62.760 |
| 9 | 1.029 | 4.116 | 67.979 | 1.029 | 4.116 | 67.979 | 1.305 | 5.219 | 67.979 |
| 10 | .895 | 3.580 | 71.559 | | | | | | |
| 11 | .850 | 3.399 | 74.958 | | | | | | |
| 12 | .779 | 3.115 | 78.073 | | | | | | |
| 13 | .706 | 2.825 | 80.899 | | | | | | |
| 14 | .642 | 2.570 | 83.469 | | | | | | |
| 15 | .588 | 2.351 | 85.819 | | | | | | |
| 16 | .515 | 2.062 | 87.881 | | | | | | |
| 17 | .496 | 1.983 | 89.864 | | | | | | |
| 18 | .454 | 1.814 | 91.678 | | | | | | |
| 19 | .413 | 1.651 | 93.329 | | | | | | |
| 20 | .365 | 1.458 | 94.787 | | | | | | |
| 21 | .330 | 1.318 | 96.106 | | | | | | |
| 22 | .280 | 1.120 | 97.226 | | | | | | |
| 23 | .240 | .961 | 98.187 | | | | | | |
| 24 | .232 | .930 | 99.116 | | | | | | |
| 25 | .221 | .884 | 100.000 | | | | | | |
| Extraction Method: Principal Component Analysis. | | | | | | | | | |

7), professional features (10-item, questions 8 to 17) responsibility (2-item, questions 18 and 19), physical, mental (4-item, questions 20 to 23) and social aspects (2-item questions 24 and 25), respectively (Table 4). To determine internal consistency, after ensuring construct validity, Cronbach's alpha coefficient in a sample of 121 midwives was 0.71 for the whole tool with 95% confidence intervals. However, this value varied from 0.73 to 0.98 in different dimensions include communications dimension ($\alpha$ = 0.73), Professional dimension ($\alpha$ = 0.85), Responsibility dimension ($\alpha$ = 0.96), Physical-Mental dimensions ($\alpha$ = 0.98), Social dimension ($\alpha$ = 0.88). In order to determine the consistency of questionnaire in the repeatability dimension, in a group of 10 midwives with 2 weeks' interval, the intra-cluster correlation coefficient (ICC) for the whole tool was 0.898. Finally, The 25-item self-reporting midwives job satisfaction tool had acceptable validity and reliability (The final Persian and English version of the Iranian midwives job satisfaction instrument and the scoring of instrument is attached in the S4–S6 Files section).

## Discussion

The purpose of this study was to design and evaluate the validation indexes of a questionnaire for measuring job satisfaction in midwives. The initial questionnaire was designed based on data extracted from qualitative study of midwives, using expert opinions and reviewing

**Table 4. The 5-factor construct, factor load of each item and the scores of Cronbach's alpha coefficient.**

| Factors name | | Items | Factor Loadings | α |
|---|---|---|---|---|
| Communications dimension | 1 | How is your relationship with the midwifery team (faculty, trainers, and faculty members)? | 0.734 | **0.73** |
| | 2 | How is your relationship with midwife colleagues at other health care centers, organizations, etc.? | 0.667 | |
| | 3 | How is your relationship with midwives working in the NGO sector? | 0.661 | |
| | 4 | How is your relationship with GPs? | 0.591 | |
| | 5 | How is your relationship with gynecologists? | 0.587 | |
| | 6 | How is your relationship with pediatricians? | 0.551 | |
| | 7 | How do you relate to those around you and ordinary people? | 0.476 | |
| Professional dimension | 1 | Does midwifery meet the economic needs of a middle-aged person? | 0.597 | **0.85** |
| | 2 | How is job security in the midwifery profession in terms of the labor market? | 0.597 | |
| | 3 | What is midwifery occupational safety? (Protecting employees from wage and salary fluctuations and eventually losing their jobs) | 0.552 | |
| | 4 | What is the flexibility of a midwife's work hours? | 0.537 | |
| | 5 | Do you think it is possible to improve practical skills in the midwifery profession? | 0.536 | |
| | 6 | How much can you use your knowledge and skills in service delivery? | 0.522 | |
| | 7 | In your opinion, how much is your workload compared to the working time during each shift? | 0.510 | |
| | 8 | How useful is the midwifery profession's usefulness in a midwife's life? | 0.481 | |
| | 9 | How much time can you upgrade your professional knowledge and skills? | 0.463 | |
| | 10 | How much can you upgrade your professional knowledge and skills in terms of available resources? | 0.417 | |
| Responsibility dimension | 1 | How are you responsible in the midwifery profession? | 0.646 | **0.96** |
| | 2 | How much do you have responsibility for your career after the end of your shift? | 0.566 | |
| Physical-Mental dimensions | 1 | How much do you want to stay in the midwifery profession in the future? | 0.554 | **0.98** |
| | 2 | How much the midwifery profession affect midwife's mental health? | 0.550 | |
| | 3 | How much is leisure time in the midwifery profession? | 0.464 | |
| | 4 | How much the midwifery profession affect midwife's physical health? | 0.419 | |
| Social dimension | 1 | How do you evaluate the social status of the midwifery profession in the community? | -0.651 | **0.88** |
| | 2 | How do you evaluate the social acceptance of the midwifery profession in the community? | 0.546 | |

existing studies in job satisfaction. After completing the validity and reliability stages, job satisfaction in midwives questionnaire was developed with 25 questions in four dimensions, which was completed by the participants. Considering that the majority of participants completed the questionnaire in 10 minutes, this tool could be easily used in screening. The findings of this study showed that, the job satisfaction in midwives questionnaire had accepted validity and reliability.

Iranian midwives job satisfaction instrument (MJSI) was finalized with five factors, including: 1) communication features, 2) professional features, 3) responsibility aspects, 4) physical-mental aspects and 5) social aspects, respectively. It is well known that, factor affecting job satisfactions varies among health professionalism, depending on the time and place [1]. In our extensive search through literature, we found no special tool to assess job satisfaction of Iranian and European midwives and studies that have examined job satisfaction in midwives have used researcher-made tools, Mueller and McCloskey Satisfaction Scale (MMSS), Minnesota questionnaire (MSQ) and the job satisfaction inventory (JDI) [18–24].

The closest study of instrument design to our study method was the Faye et al. study. Their study, aimed to develop an instrument to measure job satisfaction of medical staff including physicians, midwives, and technicians in sub-Saharan Africa with content validity, construct validity (confirmatory factor analysis) and reliability. Based on this study's result, eight dimensions, with 25 item, were identified as the main dimension of job satisfaction scale, which

included continuous education, salary and benefits, management style, tasks, work environment, workload, moral satisfaction and job stability [4]. Reliability was assessed based on internal consistency, which was good for all dimensions but moral satisfaction was lower than 0.70. Test-retest showed intra class coefficient range: 0.60 to 0.91. It was pointed out that in terms of population studied and methodology, this study is the most similar to our study. Consistent with the present study, there were similarities in the field of subscales such as professional and physical-mental aspects with the subscales of salary and benefits, workload, job stability in the mentioned tools, but in the field of subscales such as tasks, work environment, continuous education and especially Moral satisfaction and management style were not so similar. Also in the field of reliability scores, the present instrument received higher scores (0.73 to 0.98 Cronbach's alpha and ICC equal to 0.898). These differences can be due to the heterogeneity of the study population (doctors, midwives and technicians) and non-specificity for the midwifery group and the lack of item extraction in a qualitative study, different geographical location of the two studies and perhaps due to different types of structural validity in Two studies.

One of the most common tools for measuring job satisfaction is the Mueller and McCloskey Satisfaction Scale, which has been used in a variety of geographic and clinical settings such as midwives[19, 20]. The tool consists of 31 items with 5-point Likert, which are grouped into eight subscales: job satisfaction (3 items), planning (6 items), family and work balance (3 items), communication with colleagues (2 items), interaction (4 items), professional opportunities (4 items), praise and appreciation (4 items) and a sense of independence and responsibility (5 items).

The reliability of this tool has been confirmed with a Cronbach's alpha score of 0.89. In our tool, one of the extracted dimensions was the communication dimension, which is specific to the midwifery group, including items such as the individual's relationship with midwives in their main work environment and other public and private organizations, gynecologists, neonatologists, general practitioners and inspectors. Our tools also had almost identical items to the Mueller and McCloskey Satisfaction Scale in terms such as the professional dimension and the responsibility dimension. In the professional dimension of the present tool, there are questions about the use of professional skills, the possibility of upgrading practical skills, the use of skills and knowledge in providing services, workload, the opportunity to upgrade knowledge and skills, the amount of leisure, flexibility to working hours and economic security. The present tool has specific features that are not found in the Mueller and McCloskey Satisfaction Scale, such as the availability of resources and the development of professional skills and the application of the midwifery profession in life, responsibility during and after work. It is noteworthy that despite the similarities between the subscales of the present tool and the Mueller and McCloskey Satisfaction Scale, the nature of the questions in each subscale and the application of the information obtained are different from the classic Mueller and McCloskey Satisfaction Scale because the tool items were extracted from the midwife sample.

Other common tools used to measure job satisfaction are the Minnesota tool and job satisfaction inventory (JDI) which have been used to measure the job satisfaction of midwives[21, 22]. The Minnesota Job Satisfaction Questionnaire (MSQ) consists of 19 items with 5-point Likert and 6 subscales, including payment system (3 questions), type of job (4 questions), opportunities for advancement (3 questions), organizational climate (2 questions), leadership style (4 questions) and physical condition (3 questions). The reliability of the Minnesota Job Satisfaction Questionnaire using Cronbach's alpha test was .086. The job satisfaction inventory (JDI) consists of 54 items with 5-point Likert and similar subscales to the Minnesota Questionnaire that its reliability of the using Cronbach's alpha test was .094.

In our tools, there are items in the field of Physical-Mental dimensions such as job security, job stress and the desire to stay in the profession, in the social dimension such as assessing

social status and acceptance in society, as well as items in the field of professional dimension. It is emphasized again that the relative similarity of the subscales in the existing tools and the present tools, the nature of the questions that are placed in each subscale is different depending on the population that is examined especially that occupations such as medical sciences and especially midwifery have very sensitive conditions that are not comparable to other occupations. Therefore, this issue shows the value of the designed tool more than before.

Our instrument is a comprehensive and multidimensional scale with specific subscales for measuring the job satisfaction of the midwifery group. Our study includes dimensions that, due to the nature of job satisfaction, overlap with some items of available tools. But since the present study was conducted qualitatively in the first stage, which is a strength over all studies, the items of the tool are extracted from the opinions of midwives who are the main target group of this tool. Therefore, it has more credibility for measuring job satisfaction than general tools for this population group.

Of the weaknesses of the current study, only midwives employed in urban areas were studied, therefore, this is a limitation of present study. The small number of samples studied can also be considered as another limitation of this study. So, further study with larger sample size is suggested using confirmatory analysis factor. The strength of this study is that, the tool developed is the first tool that has been designed and validated exclusively with the participation of midwives only. Another strength of this study is that all steps of instrument psychometrics have been performed to confirm this instrument.

The role of midwives in the health of mother and child, that is one of the main components of primary health care, is undeniable. On the other hand, failure to pay attention to the issue of job satisfaction disrupts the organization system and causes rebellion and reduced sense of responsibility, employees' morale, and proper service delivery. Therefore, it is important to study the job satisfaction of midwives and obtain accurate information about the components of midwives' job satisfaction in order to design effective interventions with verified tool. We recommend the use of present questioner, for evaluating the job satisfaction of midwives, and also for management and research purposes.

## Conclusion

Our study showed that, Iranian midwives' job satisfaction consists of five main factors, including communications, professional features, responsibility, and physical, mental and social aspects. We recommend the use of present questioner as a valid tool for evaluating the job satisfaction of Iranian midwives.

## Supporting information

**S1 File. Questions guide in qualitative phase.**
(DOCX)

**S2 File. A minimal set of data for the study.**
(SAV)

**S3 File. Consent to participate in the project.**
(DOCX)

**S4 File. Iranian midwives job satisfaction instrument (MJSI)-English version.**
(DOCX)

**S5 File. Iranian midwives job satisfaction instrument(MJSI)-Persian version.**
(DOC)

**S6 File. Manual for scoring the MJSI.**
(DOC)

## Acknowledgments

The authors would like to thank all the midwives and those who participated in this study.

## Author Contributions

**Conceptualization:** Ashraf Direkvand-Moghadam, Mona Bahmani.

**Data curation:** Ashraf Direkvand-Moghadam, Nasrin Rashan, Mona Bahmani.

**Formal analysis:** Ashraf Direkvand-Moghadam, Nasrin Rashan, Mona Bahmani, Safoura Taheri.

**Methodology:** Ashraf Direkvand-Moghadam, Nasrin Rashan, Mona Bahmani, Safoura Taheri.

**Project administration:** Ashraf Direkvand-Moghadam, Nasrin Rashan, Mona Bahmani.

**Resources:** Ashraf Direkvand-Moghadam, Mona Bahmani.

**Software:** Ashraf Direkvand-Moghadam, Nasrin Rashan, Mona Bahmani, Safoura Taheri.

**Supervision:** Ashraf Direkvand-Moghadam, Nasrin Rashan, Mona Bahmani.

**Validation:** Ashraf Direkvand-Moghadam, Nasrin Rashan, Mona Bahmani, Safoura Taheri.

**Visualization:** Ashraf Direkvand-Moghadam, Nasrin Rashan, Mona Bahmani, Safoura Taheri.

**Writing – original draft:** Ashraf Direkvand-Moghadam, Nasrin Rashan, Mona Bahmani, Safoura Taheri.

**Writing – review & editing:** Ashraf Direkvand-Moghadam, Nasrin Rashan, Mona Bahmani, Safoura Taheri.

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
