## [Decision Letter · Decision Letter 0]

12 May 2021

PONE-D-21-09782

Development and psychometric properties of midwives job satisfaction instrument : A sequential exploratory study

PLOS ONE

Dear Dr. taheri,

Thank you for submitting your manuscript to PLOS ONE. After careful consideration, we feel that it has merit but does not fully meet PLOS ONE’s publication criteria as it currently stands. Therefore, we invite you to submit a revised version of the manuscript that addresses the points raised during the review process. I believe that following the Reviewers' suggestions greatly improve the manuscript.

We look forward to receiving your revised manuscript.

Kind regards,

Prof. Paola Gremigni, Ph.D.

Academic Editor

PLOS ONE

Journal Requirements:

2)  Thank you for stating the following financial disclosure:

 [This study was conformed by the Ethics Committee of Ilam University of Medical

Sciences .But this institution had no role in study design, data collection and analysis,

decision to publish, or preparation of the manuscript.].

3)  We note that you have indicated that data from this study are available upon request. PLOS only allows data to be available upon request if there are legal or ethical restrictions on sharing data publicly. For information on unacceptable data access restrictions, please see http://journals.plos.org/plosone/s/data-availability#loc-unacceptable-data-access-restrictions.

4)  We noticed you have some minor occurrence of overlapping text with the following previous publication(s), which needs to be addressed:

https://www.jcdr.net/ReadXMLFile.aspx?id=7981

In your revision ensure you cite all your sources (including your own works), and quote or rephrase any duplicated text outside the methods section. Further consideration is dependent on these concerns being addressed.

Reviewers' comments:

Reviewer's Responses to Questions

**Comments to the Author**

1. Is the manuscript technically sound, and do the data support the conclusions?

Reviewer #1: Yes

Reviewer #2: Yes

Reviewer #3: No

2. Has the statistical analysis been performed appropriately and rigorously? 

Reviewer #1: Yes

Reviewer #2: Yes

Reviewer #3: Yes

3. Have the authors made all data underlying the findings in their manuscript fully available?

Reviewer #1: Yes

Reviewer #2: Yes

Reviewer #3: Yes

4. Is the manuscript presented in an intelligible fashion and written in standard English?

Reviewer #1: Yes

Reviewer #2: Yes

Reviewer #3: No

5. Review Comments to the Author

Reviewer #1: Review Comments to the Author

Please use the space provided to explain your answers to the questions above. You may also include additional comments for the author, including concerns about dual publication, research ethics, or publication ethics. (Please upload your review as an attachment if it exceeds 20,000 characters) (Limit 200 to 20000 Characters)

Please see my attached review

Reviewer #2: in the abstract saying four facters while at discussion , said 5 factors... and not clear with their respective

rationale for inclusion criteria and experience at least 5 year, but at the result session you included participants who have <5 year experience.

Grammatic and spelling error were there.

Reviewer #3: Development and psychometric properties of midwives job satisfaction instrument : A sequential exploratory study

Thank you for inviting me to review this manuscript. Please see my comments as follows:

This study has some grammatical and typo errors and should be revised by an expert person in the field of English literature.

Abstract

1.Is this a study to evaluating job satisfaction of Iranian midwives? If so, please mention in the title.

2.Results: authors stated that derived four factors, but they mentioned six factors.

3.While authors designed a 58-item questionnaire, how they stated that used a 25-item questionnaire for psychometric evaluation?

Methods

1.As this study was development and psychometric evaluation of questionnaire, I am concerning about the name of “mix method study”.

2.What was the first question to start interview?

3.Who conducted the interviews? How long was the length of each interview?

4.Who were the 10 experts that evaluated the tool?

5.Who were 10 midwives that completed the questionnaire for test-re-test study?

Results

1.In the results section of abstract, authors stated that they developed four items for the questionnaire and in the main text they mentioned five factors.

2.Please report the characteristics of participants in both qualitative and quantitative parts of the study.

Discussion

1.The first and second paragraph of discussion are repetition of what authors wrote in background. Please revise them.

2.Why authors compared their designed questionnaire with those designed in African countries, and why not with European or other developed countries?

Conclusion

1.In the discussion section authors claimed that they found five factors, and here they report four factors.

References

Please use the abbreviation form of journals in all references.

6. PLOS authors have the option to publish the peer review history of their article (what does this mean?). If published, this will include your full peer review and any attached files.

Reviewer #1: No

Reviewer #2: No

Reviewer #3: **Yes: **Prof Parvin Abedi

---

## [Author Response · Author response to Decision Letter 0]

16 Jul 2021

HI,

Thank you for your valuable suggestions.

Major revision comments 

1. Abstract 

In the methodology part there are two types of study (design qualitative and crass sectional study design). What type’s method used to select the participants during crass -sectional study design? 

Applied to the manuscript. Line 30-32. In order to investigate the construct validity, a cross-sectional study was conducted with the participation of 121 midwives with Census sampling. 

Result: there is no any indication whether those four factors (communication, professional features, responsibility, and physical, mental, and social aspects) are they significantly affect the job satisfaction or variables used to assess the job satisfaction? 

Applied to the manuscript. Line 37-43. In the qualitative phase and after reviewing the existing texts and tools by the research team, a 58-item questionnaire was developed and then entered into the psychometric phase. Then, the tool was finalized with five factors, including :1) communication features, 2) professional features, 3) responsibility aspects, 4) physical-mental aspects and 5) social aspects, respectively. After the psychometric process, a specific questionnaire was developed to measure the midwives’ job satisfaction with 25 items which explained a total of 49.95% of the total variance.

Please rephrase the sentence Validity and reliability no need of used in the result part 

It was deleted.

2. Methodology: developments of job satisfaction, the word educayion is incorrect, like this there are a lot of words which are incorrect so need of deep correction.

All corrections were made.

3. Phase 2: Psychometric evaluation: 

You have been used to both qualitative and quantitative methods were used to determine the face validity. There different types of qualitative and quantitative methods, what types of study method did you apply for this study? No clear method:

Applied to the manuscript. Line 143-151. Face validity conducted with qualitative and quantitative methods. In the quantitative face validity, the level of difficulty, relevance and ambiguity of the items were investigated by face to face interview with 10 midwives (faculty member and midwife working in health centers, hospitals, and counseling centers). Then, to determine the quantitative face validity, the impact score method was used. An item with a score equal to or greater than 1.5 was retained and other items were deleted. A 5-point Likert scale was used for determining the importance of items. The classification of very important (5), important (4), relatively important (3), slightly important (2), and unimportant was considered (67). reference: 18. Creswell Jw, Plano Clark VL. Designing and Conducting, Mixed Method Research: Thousand Oaks Ca: Sage. 2011(

4. The calcification of very important (5), important (4), relatively important (3), slightly important (2), and unimportant was considered. The word calcification or classification I think you used for classification please correct it.where did you get such classification .are they prepared by yourself? If you have any source, please put the reference 

Applied to the manuscript. Line 143-151.Face validity conducted with qualitative and quantitative methods. In the quantitative face validity, the level of difficulty, relevance and ambiguity of the items were investigated by face to face interview with 10 midwives (faculty member and midwife working in health centers, hospitals, and counseling centers). Then, to determine the quantitative face validity, the impact score method was used. An item with a score equal to or greater than 1.5 was retained and other items were deleted. A 5-point Likert scale was used for determining the importance of items. The classification of very important (5), important (4), relatively important (3), slightly important (2), and unimportant was considered (67).). reference: 18. Creswell Jw, Plano Clark VL. Designing and Conducting, Mixed Method Research: Thousand Oaks Ca: Sage. 2011(

5. There is no any operational definition why didn’t you put? 

In the qualitative phase and after reviewing the existing texts and tools by the research team, a 58-item questionnaire was developed and then entered into the psychometric phase. Applied to the manuscript. Line 135-138.The midwives’ job satisfaction questionnaire consisting of five sections: 1) communication features, 2) professional features, 3) responsibility aspects, 4) physical -mental aspects and 5) social aspects.

6. Discussion 

Line 17 and 22 the reference style not correct just you are used the Vancouver reference style, please check the style. Applied to the manuscript. Line 254 and 259.they were corrected.

In a study, Faye et al. aimed to develop an instrument to measure job satisfaction of medical staff including physicians, midwives, and technicians in sub-Saharan Africa.

In another study, Bekro et al. examined 221 midwives in Addis Ababa in Ethiopia.

7 All tables are illegible. Use line to discriminate each variable. Applied to the manuscript. Line 387-433. they were corrected.

Tab 1 Demographic characteristics and other factors related to job satisfaction in midwives

Variable Category N(%)

Age (years) Less than 25 29(23)

 25-30 19(15)

 31-35 9(7)

 36-40 17(14)

 More than 40 47(38)

Marital status Single 35(30)

 Married 85(70)

Work experience (years) Less than 5 47(39)

 6-10 8(6)

 11-15 21(17)

 More than 16 44(36)

Current task Supervisor 3(2)

 Staff 10(8)

 Therapist 56(46)

 Training 8(6)

 Health service provider 25(20)

 Clinic staff 5(4)

 Gynecologist assistant 4(3)

 Administrative department 10(8)

Tab 2: Kaiser-Meyer-Olkin Measure and Bartlett's Sphericity test of Sampling Adequacy of the questioner of the job satisfaction among Iranian midwives

KMO and Bartlett's Test

Kaiser-Meyer-Olkin Measure of Sampling Adequacy. 0.640

Bartlett's Test of Sphericity Approx. Chi-Square 845.696

 df 300

 Sig. .000

Table 3: The total variance explained by factors of the job satisfaction tool among Iranian midwives

Component Initial Eigen values Extraction Sums of Squared Loadings Rotation Sums of Squared Loadings

 Total % of Variance Cumulative % Total % of Variance Cumulative % Total % of Variance Cumulative %

1 3.934 15.738 15.738 3.934 15.738 15.738 2.280 9.122 9.122

2 3.128 12.512 28.250 3.128 12.512 28.250 2.178 8.714 17.836

3 1.823 7.292 35.542 1.823 7.292 35.542 2.143 8.573 26.408

4 1.671 6.684 42.226 1.671 6.684 42.226 2.061 8.243 34.652

5 1.581 6.326 48.552 1.581 6.326 48.552 2.000 8.000 42.651

6 1.438 5.751 54.303 1.438 5.751 54.303 1.825 7.300 49.951

7 1.219 4.877 59.180 1.219 4.877 59.180 1.677 6.708 56.659

8 1.171 4.683 63.863 1.171 4.683 63.863 1.525 6.101 62.760

9 1.029 4.116 67.979 1.029 4.116 67.979 1.305 5.219 67.979

10 .895 3.580 71.559 

11 .850 3.399 74.958 

12 .779 3.115 78.073 

13 .706 2.825 80.899 

14 .642 2.570 83.469 

15 .588 2.351 85.819 

16 .515 2.062 87.881 

17 .496 1.983 89.864 

18 .454 1.814 91.678 

19 .413 1.651 93.329 

20 .365 1.458 94.787 

21 .330 1.318 96.106 

22 .280 1.120 97.226 

23 .240 .961 98.187 

24 .232 .930 99.116 

25 .221 .884 100.000 

Extraction Method: Principal Component Analysis. 

Table 4: The 5-factor construct , factor load of each item and the scores of Cronbach's alpha coefficient

Factors name Items Factor Loadings α

Communications dimension 1 How is your relationship with the midwifery team (faculty, trainers, and faculty members)? 0.734 0.73

 2 How is your relationship with midwife colleagues at other health care centers, organizations, etc.? 0.667 

 3 How is your relationship with midwives working in the NGO sector? 0.661 

 4 How is your relationship with GPs? 0.591 

 5 How is your relationship with gynecologists? 0.587 

 6 How is your relationship with pediatricians? 0.551 

 7 How do you relate to those around you and ordinary people? 0.476 

Professional dimension 1 Does midwifery meet the economic needs of a middle-aged person? 0.597 0.85

 2 How is job security in the midwifery profession in terms of the labor market? 0.597 

 3 What is midwifery occupational safety? (Protecting employees from wage and salary fluctuations and eventually losing their jobs) 0.552 

 4 What is the flexibility of a midwife's work hours? 0.537 

 5 Do you think it is possible to improve practical skills in the midwifery profession? 0.536 

 6 How much can you use your knowledge and skills in service delivery? 0.522 

 7 In your opinion, how much is your workload compared to the working time during each shift? 0.510 

 8 How useful is the midwifery profession's usefulness in a midwife's life? 0.481 

 9 How much time can you upgrade your professional knowledge and skills? 0.463 

 10 How much can you upgrade your professional knowledge and skills in terms of available resources? 0.417 

Responsibility dimension 1 How are you responsible in the midwifery profession? 0.646 0.96

 2 How much do you have responsibility for your career after the end of your shift? 0.566 

Physical-Mental dimensions 1 How much do you want to stay in the midwifery profession in the future? 0.554 0.98

 2 How much the midwifery profession affect midwife's mental health? 0.550 

 3 How much is leisure time in the midwifery profession? 0.464 

 4 How much the midwifery profession affect midwife's physical health? 0.419 

Social dimension 1 How do you evaluate the social status of the midwifery profession in the community? -0.651 0.88

 2 How do you evaluate the social acceptance of the midwifery profession in the community? 0.546 

2) Thank you for stating the following financial disclosure:

 [This study was confirmed by the Ethics Committee of Ilam University of Medical

Sciences .But this institution had no role in study design, data collection and analysis,

decision to publish, or preparation of the manuscript.].

a. Please clarify the sources of funding (financial or material support) for your study. List the grants or organizations that supported your study, including funding received from your institution.

d. If you did not receive any funding for this study, please state: “The authors received no specific funding for this work.”

Applied to the manuscript. Line 308-310.This study was confirmed by the Ethics Committee of Ilam University of Medical Sciences. The funders had no role in study design, data collection and analysis, decision to publish, or preparation of the manuscript.

3) We note that you have indicated that data from this study are available upon request. PLOS only allows data to be available upon request if there are legal or ethical restrictions on sharing data publicly. For information on unacceptable data access restrictions, please see http://journals.plos.org/plosone/s/data-availability#loc-unacceptable-data-access-restrictions.

Applied to the manuscript. Line 298-300.There are no ethical or legal restrictions on sharing a de-identified data set. The datasets used and/or analyzed during the current study are available from the corresponding author on reasonable request

4) We noticed you have some minor occurrence of overlapping text with the following previous publication(s), which needs to be addressed:

https://www.jcdr.net/ReadXMLFile.aspx?id=7981

In your revision ensure you cite all your sources (including your own works), and quote or rephrase any duplicated text outside the methods section. Further consideration is dependent on these concerns being addressed.

Applied to the manuscript. Line 97-98. It should be noted that the main researchers in this study have experience in instrument design and psychometrics [14,15].

Reference

14. Taheri S, Tavousi M, Momenimovahed Z, Direkvand-Moghadam A, Tiznobaik A, Suhrabi Z, Taghizadeh Z. Development and psychometric properties of maternal health literacy inventory in pregnancy. PLoS One. 2020 Jun 11;15(6):e0234305. 

15. Direkvand-Moghadam A, Delpisheh A, Montazeri A, Sayehmiri K. Quality of life in infertile menopausal women; Development and psychometric of an instrument. J. clin. diagn.2016;10(6): Ic01-Ic05.

Reviewers' comments:

Reviewer's Responses to Questions

Comments to the Author

1. Is the manuscript technically sound, and do the data support the conclusions?

Reviewer #1: Yes

Reviewer #2: Yes

Reviewer #3: No

2. Has the statistical analysis been performed appropriately and rigorously?

Reviewer #1: Yes

Reviewer #2: Yes

Reviewer #3: Yes

3. Have the authors made all data underlying the findings in their manuscript fully available?

Reviewer #1: Yes

Reviewer #2: Yes

Reviewer #3: Yes

4. Is the manuscript presented in an intelligible fashion and written in standard English?

Reviewer #1: Yes

Reviewer #2: Yes

Reviewer #3: No

5. Review Comments to the Author

Reviewer #1: Review Comments to the Author

Please use the space provided to explain your answers to the questions above. You may also include additional comments for the author, including concerns about dual publication, research ethics, or publication ethics. (Please upload your review as an attachment if it exceeds 20,000 characters) (Limit 200 to 20000 Characters)

Please see my attached review

Reviewer #2: in the abstract saying four facters while at discussion , said 5 factors... and not clear with their respective

Applied to the manuscript. Line 37-43.Then, the tool was finalized with five factors, including :1) communication features, 2) professional features, 3) responsibility aspects, 4) physical-mental aspects and 5) social aspects, respectively. After the psychometric process, a specific questionnaire was developed to measure the midwives’ job satisfaction with 25 items which explained a total of 49.95% of the total variance.

rationale for inclusion criteria and experience at least 5 year, but at the result session you included participants who have <5 year experience.

Applied to the manuscript. Line 120-122 and 171-173.In this study, in qualitative phase, the inclusion criteria were; being a faculty member or a midwife with at least 5 years of experience in health centers, hospitals, and counseling centers who were key informants. But in quantitative phase, in order to investigate the construct validity, a cross-sectional study was conducted with the participation of 121 midwives with census sampling. therefore, the inclusion criteria were; all faculty member and all midwife working in health centers, hospitals, and counseling centers that would like to participate in the study, with any level of work experience. 

Grammatic and spelling error were there. 

It is correct.

Reviewer #3: Development and psychometric properties of midwives job satisfaction instrument : A sequential exploratory study

Thank you for inviting me to review this manuscript. Please see my comments as follows:

This study has some grammatical and typo errors and should be revised by an expert person in the field of English literature.

Abstract

1.Is this a study to evaluating job satisfaction of Iranian midwives? If so, please mention in the title.

Applied to the manuscript. Line 1-2.Development and psychometric properties of Iranian midwives job satisfaction instrument : A sequential exploratory study

2.Results: authors stated that derived four factors, but they mentioned six factors.

Applied to the manuscript. Line 38-43.Then, the tool was finalized with five factors, including :1) communication features, 2) professional features, 3) responsibility aspects, 4) physical-mental aspects and 5) social aspects, respectively.

3.While authors designed a 58-item questionnaire, how they stated that used a 25-item questionnaire for psychometric evaluation?

Applied to the manuscript. Line 37-43.The initial questionnaire was 58 items, which reached 25 items in the validity and reliability stages by removing the items in different stages.

Methods

1.As this study was development and psychometric evaluation of questionnaire, I am concerning about the name of “mix method study”.

Applied to the manuscript. Line 103-110.This research started with a qualitative study and continued with a quantitative study. This method was introduced by Creswell and Plano Clark as one of the five main types of mix method studies (16). The sequential exploratory study is divided into two types of theory design and instrument design. The present study is a tool design study (17). To build a tool, we need to understand the concept of job satisfaction by using the experience of participants. Having a conceptual framework is the first step in designing a tool. Therefore, the qualitative part of the study was designed and implemented to design the tool, and then its psychometric property was confirmed in a quantitative study.

16. Creswell JW, Plano Clark VL. Designing and Conducting Mixed Methods Research. 2007.

17. Creswell J W, Clark V L. P. Designing and Conducting Mixed Methods Research, SAGE.2014.

2.What was the first question to start interview?

Applied to the manuscript. Line 124-128.The first question to start interview was: “What comes to mind when you hear the word job satisfaction? As the study progressed, additional questions or new questions were asked from the interviewees. Exploratory questions such as: "What do you mean?" or "If you can please explain more" or Please explain to me? or Please tell me about your experience were also used in the interviews as needed

3.Who conducted the interviews? How long was the length of each interview?

Applied to the manuscript. Line 129-131.The interview was conducted by the researcher. The duration of each interview was between 45 to110 minutes, depending on the amount of information, participation, and cooperation of the participants.

4.Who were the 10 experts that evaluated the tool?

Applied to the manuscript. Line 158-159.In order to evaluate the qualitative content validity, 10 experts in gynecology, midwifery, reproductive health, maternal and child health, health education participated.

5.Who were 10 midwives that completed the questionnaire for test-re-test study?

Applied to the manuscript. Line 181.To determine the stability of the questionnaire, 10 midwives (faculty member and midwife working in health centers, hospitals, and counseling centers) participated. They were asked to complete the final questionnaire twice, at an interval of 2 weeks.

Results

1.In the results section of abstract, authors stated that they developed four items for the questionnaire and in the main text they mentioned five factors.

Applied to the manuscript. Line 37-43. Then, the tool was finalized with five factors, including :1) communication features, 2) professional features, 3) responsibility aspects, 4) physical-mental aspects and 5) social aspects, respectively. After the psychometric process, a specific questionnaire was developed to measure the midwives’ job satisfaction with 25 items which explained a total of 49.95% of the total variance.

2.Please report the characteristics of participants in both qualitative and quantitative parts of the study.

Applied to the manuscript. Line 18-191.In qualitative phase, in total, 10 expert person aged 27- 45 years participated in this study. The lowest work experience was 5 years and the highest work experience was 26 years. The participants’ current tasks of the participants were included two supervisors, two staff, two Training, two health service provider, one gynecologist assistant and one clinic staff.

Applied to the manuscript. Line 202.In quantitative phase, in construct validity, Demographic data for the sample are presented in table 1.

Tab 1 Demographic characteristics and other factors related to job satisfaction in midwives

Variable N(%)

Age (years) Less than 25 29(23)

 25-30 19(15)

 31-35 9(7)

 36-40 17(14)

 More than 40 47(38)

Marital status Single 35(30)

 Married 85(70)

Work experience (years) Less than 5 47(39)

 6-10 8(6)

 11-15 21(17)

 More than 16 44(36)

Current task Supervisor 3(2)

 Staff 10(8)

 Therapist 56(46)

 Training 8(6)

 Health service provider 25(20)

 Clinic staff 5(4)

 Gynecologist assistant 4(3)

 Administrative department 10(8)

Discussion

1.The first and second paragraph of discussion are repetition of what authors wrote in background. Please revise them.

It was correct.

Applied to the manuscript. Line 225-233.The purpose of this study was to design and evaluate the validation indexes of a questionnaire for measuring job satisfaction in midwives. The initial questionnaire was designed based on data extracted from qualitative study of midwives, using expert opinions and reviewing existing studies in job satisfaction. After completing the validity and reliability stages, job satisfaction in midwives questionnaire was developed with 25 questions in four dimensions, which was completed by the participants. Considering that the majority of participants completed the questionnaire in 10 minutes, this tool could be easily used in screening. The findings of this study showed that, the job satisfaction in midwife’s questionnaire had accepted validity and reliability.

2.Why authors compared their designed questionnaire with those designed in African countries, and why not with European or other developed countries?

Applied to the manuscript. Line 247-250.In our extensive search through literature, we only found two studies that have been done to design and psychometric job satisfaction tools in Iranian physicians and nurses [14,15], and no tool was found to assess job satisfaction of Iranian and European midwives.

Conclusion

1.In the discussion section authors claimed that they found five factors, and here they report four factors.

Applied to the manuscript. Line 282-285.Our study showed that, Iranian midwives’ job satisfaction consists of five main factors, including communications, professional features, responsibility, and physical, mental and social aspects. We recommend the use of present questioner as a valid tool for evaluating the job satisfaction of Iranian midwives.

References

Please use the abbreviation form of journals in all references. Applied to the manuscript. Line 327-384.

References

1. Khavayet F, Tahery N, Alizadeh Ahvazi M, Tabnak A. A Survey of Job Satisfaction among Midwives Working in Hospitals. JMRH. 2018;6(1):1186-92.

2. Lu AC, Gursoy D. Impact of job burnout on satisfaction and turnover intention: do generational differences matter?. JHTR. 2016 Feb;40(2):210-35.

3. Liu D, Mitchell TR, Lee TW, Holtom BC, Hinkin TR. When employees are out of step with coworkers: How job satisfaction trajectory and dispersion influence individual-and unit-level voluntary turnover. Acad. Manag. Ann.. 2012 Dec;55(6):1360-80.

4. Faye A, Fournier P, Diop I, Philibert A, Morestin F, A D. Developing a tool to measure satisfaction among health professionals in sub-Saharan Africa. Hum Resour Health. 2013 11(30).

5. Fortney L, Luchterhand C, Zakletskaia L, Zgierska A, Rakel D. Abbreviated mindfulness intervention for job satisfaction, quality of life, and compassion in primary care clinicians: a pilot study. Ann. Fam. Med.. 2013 Sep 1;11(5):412-20.

6. Bekru ET, Cherie A, Anjulo AA. Job satisfaction and determinant factors among midwives working at health facilities in Addis Ababa city, Ethiopia. PLoS One. 2017 Feb 17;12(2):e0172397.

7. Lippincott Williams, Wilkins . Midwives could prevent two-thirds of maternal and newborn deaths worldwide. AJN, Am J Nurs. 2014;114(9):18.

8. Singh JK, Jain M. A Study of employee’s job satisfaction and its impact on their performance. Journal of Indian research. 2013 Oct;1(4):105-11.

9. Kumar R, Ahmed J, Shaikh BT, Hafeez R, Hafeez A. Job satisfaction among public health professionals working in public sector: a cross sectional study from Pakistan. Hum Resour Health. 2013; 11:2.

10. Pinar SE, Ucuk S, Aksoy OD, Yurtsal ZB, Cesur B, et al. Job Satisfaction and Motivation Levels of Midwives/Nurses Working in Family Health Centres: A Survey from Turkey. Int. J. Caring Sci..2017; 10: 802-812 .11. Hadizadeh Talasaz Z, Nourani Saadoldin Sh, Shakeri MT. The Relationship between Job Satisfaction and Job Performance among Midwives Working in Healthcare Centers of Mashhad, Iran. JMRH. 2014;2(3): 157-164

12. Uchmanowicz I, Manulik S, Lomper K, Rozensztrauch A, Zborowska A, Kolasińska J, Rosińczuk J. Life satisfaction, job satisfaction, life orientation and occupational burnout among nurses and midwives in medical institutions in Poland: a cross-sectional study. BMJ Open. 2019 Jan 1;9(1):e024296.

13. Bloxsome D, Ireson D, Doleman G, Bayes S. Factors associated with midwives’ job satisfaction and intention to stay in the profession: An integrative review. J. Clin. Nurs.. 2019 Feb;28(3-4):386-99.

14. Taheri S, Tavousi M, Momenimovahed Z, Direkvand-Moghadam A, Tiznobaik A, Suhrabi Z, Taghizadeh Z. Development and psychometric properties of maternal health literacy inventory in pregnancy. PLoS One. 2020 Jun 11;15(6):e0234305. 

15. Direkvand-Moghadam A, Delpisheh A, Montazeri A, Sayehmiri K. Quality of life in infertile menopausal women; Development and psychometric of an instrument. J. clin. diagn.2016;10(6): Ic01-Ic05.

16. Creswell JW, Plano Clark VL. Designing and Conducting Mixed Methods Research. 2007.

17. Creswell J W, Clark V L. P. Designing and Conducting Mixed Methods Research, SAGE.2014.

18.Creswell Jw, Plano Clark VL. Designing and Conducting, Mixed Method Research: Thousand Oaks Ca: Sage. 201119. Soleimani MA, Sharif SP, Yaghoobzadeh A, Panarello B. Psychometric evaluation of the moral distress scale–revised among Iranian nurses. Nurs. Ethics. 2019 Jun;26(4):1226-42.

20. Salaree M, Zareiyan A, Ebadi A. Development and Psychometric Properties of the Military Nurses' Job Burnout Factors Questionnaire. J Mil Med. 2019;20(6):645-54.

21. Wilson B, Squires MA, Widger K, Cranley L, Tourangeau AN. Job satisfaction among a multigenerational nursing workforce. J. Nurs. Manag.. 2008 Sep;16(6):716-23.

6. PLOS authors have the option to publish the peer review history of their article (what does this mean?). If published, this will include your full peer review and any attached files.

Do you want your identity to be public for this peer review? For information about this choice, including consent withdrawal, please see our Privacy Policy.

Reviewer #1: No

Reviewer #2: No

Reviewer #3: Yes: Prof Parvin Abedi

---

## [Decision Letter · Decision Letter 1]

24 Sep 2021

PONE-D-21-09782R1Development and psychometric properties of Iranian midwives job satisfaction instrument ( MJSI) : A sequential exploratory studyPLOS ONE

Dear Dr. taheri,

Thank you for submitting your manuscript to PLOS ONE. After careful consideration, we feel that it has merit but does not fully meet PLOS ONE’s publication criteria as it currently stands. Therefore, we invite you to submit a revised version of the manuscript that addresses the points raised during the review process. Although the previous modification of the manuscript improved it, there are still minor revisions required  by Reviewers in this second round. I believe that following such recommendations can further improve your contribution.

We look forward to receiving your revised manuscript.

Kind regards,

Prof. Paola Gremigni, Ph.D.

Academic Editor

PLOS ONE

Journal Requirements:

Reviewers' comments:

Reviewer's Responses to Questions

**Comments to the Author**

1. If the authors have adequately addressed your comments raised in a previous round of review and you feel that this manuscript is now acceptable for publication, you may indicate that here to bypass the “Comments to the Author” section, enter your conflict of interest statement in the “Confidential to Editor” section, and submit your "Accept" recommendation.

Reviewer #2: All comments have been addressed

Reviewer #4: (No Response)

Reviewer #5: All comments have been addressed

2. Is the manuscript technically sound, and do the data support the conclusions?

Reviewer #2: Partly

Reviewer #4: Yes

Reviewer #5: Yes

3. Has the statistical analysis been performed appropriately and rigorously? 

Reviewer #2: Yes

Reviewer #4: Yes

Reviewer #5: Yes

4. Have the authors made all data underlying the findings in their manuscript fully available?

Reviewer #2: Yes

Reviewer #4: No

Reviewer #5: Yes

5. Is the manuscript presented in an intelligible fashion and written in standard English?

Reviewer #2: Yes

Reviewer #4: Yes

Reviewer #5: Yes

6. Review Comments to the Author

Reviewer #2: Thanks for invitation and giving extra time.

in th e Abstract some spelling error...like Crass sectional to be cross sectional.

Reviewer #4: I want to thanks for getting this chance to review your document. I have got it well written and interesting topic. However, it is better to consider the following issues.

• It is better to take the last sentence of the first paragraph to the end of background section.

• In phase I: why being a faculty member or a midwife with at least 5 years of experience included? Why 5 year is selected? Why not more or less?

• Ethical statement is described in two place, in method section and in declaration section. Please avoid one of it based on journal format.

• The discussion section is not in line with objective and findings. The second paragraph of discussion is typically background. The study used in the fourth paragraph of the discussion has no the same objective with your study. Therefore, it is impossible comparing these two study. the discussion have no justification for the similarity and difference occurred with other studies.

Overall, the discussion is somewhat seems poor, try two modify it.

• The sum of participants in table 1is not 121. Some are 120 some are 121. Why this difference occurred?

Reviewer #5: Thank you so much o give me an opprtunity ti re-review the study. the authors have perfomed the corrections carefully.

thre are only two comments: 1) I think it is better to report alpha cronbach for each domain not total.

2) it ould be good if the authors put the persian of questionre as supplemntry file.

Good luck

7. PLOS authors have the option to publish the peer review history of their article (what does this mean?). If published, this will include your full peer review and any attached files.

Reviewer #2: **Yes: **Tiruset Gelaw(BSc, MSc in clinical midwifery)

Reviewer #4: No

Reviewer #5: **Yes: **Dr Leila Amiri-Farahani

---

## [Author Response · Author response to Decision Letter 1]

2 Dec 2021

_HI,

Thank you for your valuable suggestions.

6. Review Comments to the Author

Reviewer #2: Thanks for invitation and giving extra time.

in th e Abstract some spelling error...like Crass sectional to be cross sectional. It was corrected. Abstract section. Line 31 and 32.

Reviewer #4: I want to thanks for getting this chance to review your document. I have got it well written and interesting topic. However, it is better to consider the following issues.

• It is better to take the last sentence of the first paragraph to the end of background section. It was corrected. background section. Line 56.

• In phase I: why being a faculty member or a midwife with at least 5 years of experience included? Why 5 year is selected? Why not more or less? In the explanation it should be said that According to the work experience of researchers and also consulting with experts in this field and according to the specific issue of job satisfaction in order to obtain the best results from the qualitative section of key informants, the inclusion criterion of at least 5 years of work experience was considered.

• Ethical statement is described in two place, in method section and in declaration section. Please avoid one of it based on journal format. It was corrected. Acknowledgments section. line 349.

• The discussion section is not in line with objective and findings. The second paragraph of discussion is typically background. The study used in the fourth paragraph of the discussion has no the same objective with your study. Therefore, it is impossible comparing these two study. the discussion have no justification for the similarity and difference occurred with other studies.

Overall, the discussion is somewhat seems poor, try two modify it. 

Dear Reviewer

Hi, we tried very hard to write a worthy discussion, even though there was no tool for job satisfaction in the midwifery group and only one manuscript in the medical sciences in the field of tools. We hope it is satisfactory. It was corrected. Discussion section. Line 231-319. 

• The sum of participants in table 1is not 121. Some are 120 some are 121. Why this difference occurred? . It was corrected. Table 1. line 405.

Reviewer #5: Thank you so much o give me an opprtunity ti re-review the study. the authors have perfomed the corrections carefully.

thre are only two comments: 1) I think it is better to report alpha cronbach for each domain not total. It was corrected. result section. line 214-216.

2) it ould be good if the authors put the persian of questionre as supplemntry file. Applied to the manuscript. It was placed in the supplementary part.

Good luck

---

## [Editor Report · Decision Letter 2]

4 Jan 2022

Development and psychometric properties of Iranian midwives job satisfaction instrument ( MJSI) : A sequential exploratory study

PONE-D-21-09782R2

Dear Dr. taheri,

We’re pleased to inform you that your manuscript has been judged scientifically suitable for publication and will be formally accepted for publication once it meets all outstanding technical requirements.

Kind regards,

Paola Gremigni, Ph.D.

Academic Editor

PLOS ONE

---

## [Editor Report · Acceptance letter]

14 Jan 2022

PONE-D-21-09782R2 

Development and psychometric properties of Iranian midwives job satisfaction instrument( MJSI) : A sequential exploratory study 

Dear Dr. Taheri:

I'm pleased to inform you that your manuscript has been deemed suitable for publication in PLOS ONE. Congratulations! Your manuscript is now with our production department. 

Kind regards, 

on behalf of

Prof. Paola Gremigni 

Academic Editor

PLOS ONE